# The Usefulness of Sexual Behaviour Assessment at the Beginning of Service to Predict the Suitability of Boars for Artificial Insemination

**DOI:** 10.3390/ani11123341

**Published:** 2021-11-23

**Authors:** Stanisław Kondracki, Maria Iwanina, Anna Wysokińska, Dorota Banaszewska, Władysław Kordan, Leyland Fraser, Katarzyna Rymuza, Krzysztof Górski

**Affiliations:** 1Institute of Animal Production and Fisheries, Faculty of Agrobioengineering and Animal Husbandry, Siedlce University of Natural Sciences and Humanities, Prusa 14, 08-110 Siedlce, Poland; stanislaw.kondracki@uph.edu.pl (S.K.); maria.iwanina@uph.edu.pl (M.I.); anna.wysokinska@uph.edu.pl (A.W.); dorota.banaszewska@uph.edu.pl (D.B.); 2Department of Animal Biochemistry and Biotechnology, University of Warmia and Mazury in Olsztyn, Oczapowskiego 5, 10-719 Olsztyn, Poland; wladyslaw.kordan@uwm.edu.pl (W.K.); fraser@uwm.edu.pl (L.F.); 3Institute of Agriculture and Horticulture, Faculty of Agrobioengineering and Animal Husbandry, Siedlce University of Natural Sciences and Humanities, Prusa 14, 08-110 Siedlce, Poland; katarzyna.rymuza@uph.edu.pl

**Keywords:** boar, ejaculate, sexual activity

## Abstract

**Simple Summary:**

Boars producing ejaculates of high quality and shortest duration are particularly valuable; this aspect can be the primary factor in inbreeding development and improvement of swine utilization. It is important to develop objective ways to describe the sexual activity of various breeds of boars and its application and effectiveness in conducting artificial insemination. The results of this study indicate that the time from entering the arena until achieving an erection, which is assessed at the beginning of a boar’s breeding utilization, may be used to predict a boar’s future libido. There was an association between the level of libido and ejaculate characteristics. Boars requiring the most time to begin ejaculation produce ejaculates with a higher sperm concentration and number.

**Abstract:**

Parameters of sexual activity were determined in 49 young boars used for artificial insemination, four times at three-month intervals. The parameters included the time from entering the arena until mounting the phantom; the time from mounting the phantom until achieving erection; the time from achieving full erection until the start of ejaculation; duration of ejaculation; and the number of times the boar mounted the phantom. Characteristics of the ejaculates were also assessed. The libido parameter associated with the greatest efficacy of artificial insemination was the effectiveness of artificial insemination service, the time from entering the arena until the start of ejaculation. The significance of this trait for predicting ejaculation performance was analysed. The libido characteristics were classified into three categories: boars with a short reaction time to the phantom, boars with an intermediate reaction time, and boars with a long reaction time. For these groups, the characteristics of ejaculates collected at the start of the period during which ejaculates were collected and after three, six and nine months were determined. The sexual experience of boars was not associated with the expression of sexual behaviour because young boars during their first three months of ejaculate collections required less time to initiate ejaculation. The ejaculates with the greatest utility were obtained after six months of service. These ejaculates had the largest volume (255.22 mL), and the most insemination doses could be prepared from these ejaculates. On average, more than 23 insemination doses were prepared from ejaculates collected after six months of semen collections, which is about four doses more than from ejaculates collected at the start of artificial insemination service (*p* < 0.01).The time from entering the arena to beginning ejaculation can be used to predict a boar’s future libido. A relationship was shown between the level of libido and ejaculate characteristics. The ejaculates of the boars which needed the longest time to begin ejaculation at the start of semen collections had the greatest sperm concentration and number. In group 3, the boars’ejaculates contained about 6–9 × 10^9^ more sperm than the ejaculates of boars from group 1. After six months of the experimental period, the difference was nearly 15 × 10^9^ sperm (*p* < 0.05), and after nine months, it exceeded 22 × 10^9^ sperm (*p* < 0.01).

## 1. Introduction

The profitability of animal production depends on reproductive success. Breeding efficiency is particularly important in polytocous species such as the domestic pig, and exploitation of the potential fertility of sows largely depends on the reproductive capacity of the boar [1]. Boars used for breeding should have a high level of sexual activity and produce a large quantity of high-quality semen [2]. An important factor for the effectiveness of artificial insemination is the sexual activity of male pigs. The amount of sexual activity is determined by the complex interaction between the body of the male and the external environment [3]. Knowledge of the mechanisms regulating sexual behaviour is important for optimizing the reproductive performance of boars [4]. The current state of knowledge of the importance of libido for the ejaculatory performance of boars is limited in comparison to the knowledge of the physiology of semen production [5]. In practice, boars are selected for artificial insemination mainly according to the criterion of breeding value and market demand for boars of a given breed or cross-breeding variant. Less is known, however, of a boar’s suitability for future use for artificial insemination. There are few studies devoted to establishing criteria for estimating the future ejaculatory performance of boars or the quality of the sperm produced during subsequent ejaculations [6]. It is also difficult to predict the amount of libido that a young boar will have in the future. No methods have been developed for predicting the suitability of young boars for artificial insemination services. The lack of such a method leads to the early culling of many individuals or maintaining them despite their poor performance. The decision to cull and replace a boar is often not made until after it has been determined that his semen cannot be used effectively for conducting artificial inseminations. This leads to considerable economic losses, not only among owners of boars but also in the breeding and production of herds where owners make use of artificial insemination.

A greater sex drive has a beneficial effect on ejaculation. It facilitates the transport of sperm through the ejaculatory ducts, which stimulates ejaculation performance. The boar’s libido is important for the required frequency of semen collection, without complications [4]. According to some authors, boars with relatively greater libidos produce ejaculates with more favourable values for parameters than boars with lesser sexual activity [7,8]. By stimulating the sexual behaviour of males, it is possible to affect the quality of their ejaculates [5,9]. The results of some studies indicate that libido traits can be determined in young boars [10]. It is possible that certain parameters of the sexual activity of young males can be used as a procedure to predict suitability for boar selections when boars are used for artificial insemination, even after sexual maturity.

In the present study, we attempted to determine the time needed for young boars to begin ejaculation at the start of the period when boars ejaculate, as such measurements are being collected as criteria for estimating their suitability in the utilization of semen for artificial insemination.

## 2. Materials and Methods

The study comprises accepted methods and standard operating procedures for boar semen processing. The centres in this research complied with the Council Directive, 2008/120/EC outlining minimum standards for the protection of pigs and directive, as well as with 2010/63/EU of the European Parliament and of the Council of 22 September 2010 on the protection of animals used for scientific purposes. Animal experiments were conducted in accordance with the guidelines set out by the Polish Local Ethics Committee, Warsaw, Poland (Number: 51/2015) and by the Polish Laboratory Animal Science Association (Numbers 3235/2015 and 4466/2017).

The study was conducted on 49 boars used at commercial insemination stations. Young boars at the start of their use for breeding, at the age of 7–8 months, were selected for the study. The boars were kept under the same controlled environmental conditions.They were kept on litter in individual pens with an area of 6.3 m^2^, on a concrete floor with thermal and moisture isolation. The boars selected for the experiments were clinically healthy and had normal locomotor function. Electronically controlled mechanical ventilation was used in the pig houses. All the boars were housed in pens with adjacent paddocks. Boars were fed individually with a pelleted complete feed. Feed was provided twice a day, at 6:30 a.m. and 1:00 p.m. The boars had constant access to drinking water from the nipples of watering devices.

The study consisted of two stages:

Stage I. Identification of the most important parameter of sexual activity for the effectiveness of selecting boars for use in artificial insemination.

For each boar selected for the study, sexual activity parameters were determined four times. Libido was tested at three-month intervals, as follows:At the start of the period during which there were semen collections (at the age of 7–8 months);Three months after the start of semen collections (at the age of 10–11 months);Six months after the start of semen collections (at the age of 13–14 months);Nine months after the start of semen collections (at the age of 16–17 months).

Semen collections were carried out at the site of the boar semen collection centre. The boars’ sexual activity was assessed, based on the time needed to induce successive copulatory reflexes during the period of semen collection. Boars were not exposed to oestrus sows during semen collection.

The following parameters of the boars’ sexual activity were measured:Time from entering the arena until mounting the phantom;Time from mounting the phantom until achieving an erection;Time from achieving full erection until the start of ejaculation;Duration of ejaculation;Number of times the boar needed to mount the phantom in order to ejaculate.

The following were calculated as well:Time from entering the arena until achieving an erection;Time from entering the arena until the start of ejaculation;Duration of copulation, i.e., the total time from successfully mounting the phantom until dismounting the phantom after ejaculation was completed.

The time required to induce individual sexual reflexes and their duration was determined using a stopwatch, to within one second.

During testing of the sexual activity of the boars, ejaculates were collected in the morning twice per week, using the “gloved-hand” technique [11] and placing the semen into a sterile plastic bag placed in a thermally insulated vessel sealed with a single-use viscose filter enabling separation of the gel fraction. The same trained technicians performed all semen collections.

The freshly collected ejaculates during libido measurements underwent standard laboratory assessment, including the following ejaculate traits:Ejaculate volume (mL);Sperm concentration in the ejaculate (×10^3^/mm^3^);Sperm motility, as the percentage of sperm with progressive movement (%);Number of motile sperm in the ejaculate (×10^9^).

In addition, the number of insemination doses that could be prepared from one ejaculate was calculated.

The ejaculate traits were determined according to the methods used at Polish sow insemination stations. Ejaculate volume was determined after separating the gel fraction. The sperm concentration in the ejaculate was determined using the photometric method using a Spermacue photometer (Minitube, GmbH, Tiefenbach, Germany). This method involves quantifying the intensity of light passing through a microcuvette containing an undiluted semen sample. The length of the light path was 0.7 μm. Sperm motility was determined by microscope examination. The percentage of spermatozoa having normal progressive movement in the total number of sperm visible in the field of view of the microscope was estimated in a drop of fresh semen under 200× magnification. The total number of motile sperm in the ejaculate and the number of insemination doses obtained from a single ejaculate were calculated using WINSUL software (v.6.3.5; Gogosystem, Poland).

Statistical analysis of the results was performed. Changes in the sexual activity of boars and the quality of their ejaculates were analysed over the course of their artificial insemination service. The data were analysed using the following mathematical model:Y*_ij_* = μ + a*_i_* + e*_ij_*
where Y*_ij_* is the value of the trait; μ is the population mean; a*_i_* is the effect of duration of artificial insemination service; e*_ij_* is the error.

The significance of differences between groups was determined using Tukey’s test.

Stage II. Verification of the usefulness of the parameter of sexual activity selected for predicting the artificial insemination performance of boars.

Based on the results of the first stage of the study, the most important parameter of sexual activity for determining the efficacy of semen used for artificial insemination of boars was identified. The parameter chosen was the time from entering the arena until the start of ejaculation. This parameter could be used to determine the duration of time before the boar’s response to the phantom; it was highly variable, with substantial differences among groups of boars. The significance of this parameter for predicting the ejaculatory performance of boars was analysed in detail. For this purpose, the empirical material was grouped according to the sexual activity of the boar at the start of the period during which there were semen collections with there being classifications into three groups:

Group 1—boars with a short reaction time to the phantom;

Group 2—boars with an intermediate reaction time;

Group 3—boars with a long reaction time.

For these groups, the average time from entering the arena until the start of ejaculation was calculated, and values for ejaculate parameters were determined, using the same methodology as in the first stage of the study. Statistical analysis of the results was performed using the following mathematical model:Y*_ij_* = μ + a*_i_* + e*_ij_*
where Y*_ij_* is the value of the trait; μ is the population mean; a*_i_* is the reaction to the phantom (time from entering the arena until the start of ejaculation); e*_ij_* is the error.

The determination as to when there were mean differences was made using Tukey’s test.

## 3. Results

### 3.1. Stage I

The data for the determination of the sexual activity of boars, taking into account the period when semen collections occurred, are presented in Table 1.

The data indicate that the sexual activity of boars, measured as the time needed to induce individual copulatory reflexes, did not change during the period over which semen collections occurred. There were no statistically confirmed differences in the time needed to induce individual copulatory reflexes at the start of the period over which semen collections occurred or after three, six and nine months of semen collections. This is probably due to the large amount of variation in the parameters tested. However, certain trends can be observed in the sexual activity of the boars. Boars where the initial semen collection occurred while mounting a phantom required a relatively long time to successfully mount the phantom, and they mounted it more times before ejaculating. After three months of semen collection, the time needed to successfully mount the phantom was about 20 s less than subsequent periods of semen collection duration to successfully mount the phantom. Similar tendencies can be observed in the case of the time required to induce other copulatory reflexes, such as the time from mounting the phantom until achieving an erection, the time from entering the arena until achieving an erection, and the time from entering the arena until the start of ejaculation. It cannot be discounted that the slightly longer time needed to induce copulatory reflexes in young boars is due to their lack of experience. After three months of semen collections, the boars were well acquainted with the semen collection process and more quickly displayed the copulatory reflexes leading to ejaculation. As the duration of the period over which semen collections occurred increased to longer than six and nine months, the boars’ reaction time to stimuli associated with semen collection gradually increased, which may be due to their increasing experience.

Data in Table 2 show that for ejaculates obtained from boars at the initial semen collection and after three, six and nine months of semen collection.

The data indicate that the ejaculate traits changed during the period of the experiment when semen collections were occurring. The ejaculates with the most utility were obtained after six months of semen collection. These ejaculates had the largest volume (255.22 mL), allowing for the most insemination doses to be prepared. On average, more than 23 insemination doses were prepared from ejaculates collected after six months of semen collection, which is about four doses more than from ejaculates collected at the start of artificial insemination service (*p* < 0.01). These ejaculates also had the largest number of sperm.

During the experimental period that ejaculates were being collected, the total sperm number per ejaculate and the number of insemination doses per ejaculate also increased. At the start of the semen collection period, the boars produced ejaculates containing 58.18 × 10^9^ sperm on average. The sperm count in the ejaculate increased by 4.17 × 10^9^ after three months and by 6.24 × 10^9^ after six months of semen collections. These changes were not confirmed statistically, but this trend still indicates there was a larger number of sperm in the ejaculate as the experimental period advanced and sexual development occurred. This results in changes in the number of insemination doses prepared from the ejaculate. Only 19.1 insemination doses were obtained from the ejaculates of the youngest boars, but the number of doses increased by 2.82 after three months of semen collection and by 3.96 after six months (*p* < 0.01).

### 3.2. Stage II

During Stage II of the study, the usefulness of the selected sexual activity parameter (the time needed to begin ejaculation) for predicting the utility of the semen of a specific boar for artificial insemination was determined.

Table 3 shows the measurement results of the sexual activity of boars’ time required to begin ejaculation.

The data indicate that the sexual activity of the boars at the start of the period of semen collection, measured as the time from entering the arena until the start of ejaculation, continued to be associated with the sexual activity of these boars after three, six and nine months of service. Boars needing the most time to begin ejaculation at the start of the period of semen collection also needed the most time after three, six and nine months of the semen collection period. These trends were pronounced, but the significance of the differences was not statistically confirmed. It is worth noting that in the tests conducted after three, six and nine months of semen collections, there was a marked increase in variation in the length of time needed to begin ejaculation. This indicates that individuals may develop or lose the traits of natural sexual behaviour observed at the initial time of semen collection in varying degrees. As indicated by data in Table 3, the boars that needed the longest time to begin ejaculation also had the longest ejaculation time and the longest total copulation time and also mounted the phantom the most times before achieving ejaculation. However, this was only confirmed statistically at the start of the experimental period when ejaculates were initially being collected. In the measurements conducted after three, six, and nine months, this pattern was lost.

Data in Table 4 are the values for the ejaculate traits of boars depending on the time needed to begin ejaculation at the interval start over which semen collections occurred.

The data indicate that the libido of boars, measured as the time from entering the arena until the start of ejaculation, is associated with certain ejaculate traits. The ejaculates of the boars requiring the longest time to begin ejaculation at the start of semen collection had the greatest sperm concentration and number. This tendency was pronounced, although the differences were confirmed statistically only for ejaculates collected after six months of service (sperm concentration) and after six and nine months (sperm number). It should be noted that these differences were much smaller for ejaculates collected during the initial portion of the experimental period and after three months of this period than for ejaculates obtained after six and nine months. In the group of boars that needed more time to ejaculate (group 3), at the start of the interval during which ejaculates were collected and after three months the ejaculates contained about 6–9 × 10^9^ more sperm than the ejaculates of boars from group 1. After six months of the experimental period, the difference was nearly 15 × 10^9^ (*p* < 0.05), and after nine months it exceeded 22 × 10^9^ sperm (*p* < 0.01). Considering that an insemination dose contains about 3 × 10^9^ sperm, this means that two to three more semen doses for artificial insemination can be obtained in practice from the ejaculates of boars beginning ejaculation later in the first few months of the experimental period, five more doses after six months of semen collection, and there can be seven more doses after nine months, compared to the ejaculates of boars that begin ejaculation earlier.

## 4. Discussion

The data presented in Table 1 indicate that the sexual experience and characteristics of boars used for artificial insemination are not consistent with the extent of the expression of sexual behaviour. Young boars in their first three months of semen collection required less time to begin ejaculation. Confirmation of this observation can be found in studies on male rodents [12,13,14], where results indicated that experienced male rats need more time both to mount the female [13] and to begin ejaculation [14]. The negative effect of sexual experience has also been described where there have been some studies in boars. Young boars that are less than 24 months of age required less time to mount the phantom and to begin ejaculation than boars over 36 months of age [15]. Kondracki et al. [8] reported that young boars during their first three months of ejaculate collections achieve erection and ejaculation more quickly than older boars. These results are consistent with those obtained in the present study. However, the considerable variation in the reaction time to the phantom must be taken into account. 

High variability in libido parameters was clearly evident in the present study, as well as in most of the previous studies cited in this manuscript. This variation may be due to multiple factors. Breed has an effect on the expression of male libido [16,17,18,19,20]. According to Wysokińska and Kondracki [17], boars of the Duroc breed need less time to begin ejaculation than Pietrain boars and two-breed crosses. There are studies indicating that male libido may be affected by the season of the year [21,22,23], as well as by muscularity and growth rate [24]. However, the main source of variation seems to be the boar’s individual semen being collected when mounting a phantom and its ability to respond to the phantom. Data presented by Szostak and Sarzyńska [7] indicate that the time needed to begin ejaculation has a wide range from 1.99 to 14.65 min, depending on the breed of boar. Savić and Petrović [23] reported a relatively shorter time needed for boars to begin ejaculation, ranging from 2 to 7 min (120–420 s) depending on the breed and day length. The results of the present study are similar to those reported by Kondracki et al. [8], ranging from 245 to 289 s, depending on the length of the ejaculation period.

The extent of libido may be influenced by a boar’s age [8,25,26]. Savić et al. [24] described how the initiation of semen collections at too young an age in boars has negative effects on the libido of these boars. In the present study, young boars during the first months of the experimental period needed less time to ejaculate. The libido of boars depends on the concentration of testosterone circulating in the blood [27]. Faster inducement of copulatory reflexes in the first months of the period during which ejaculates were being collected may be linked to its testosterone concentration. Allrich et al. [28] observed the maximum testosterone concentration in the blood of boars at the age of 7–8 months, i.e., at the age when there was the initiation of the use of these boars for breeding. Based on results from this previous study, a boar’s testosterone concentration is greatest between 190 and 220 days of life (15.75–15.76 ng/mL), after which its secretion decreases. Similar results were reported by Weiler and Claus [29], where the testosterone concentration in the blood of boars increased up to the age of 32–35 weeks (8–9 months) and then decreased, and that the scale and rate of these changes depend on the breed of boar. These data are consistent with the results of the present study, which indicate that copulatory reflexes are induced most quickly in young boars, but not until after three months of the period during which ejaculates were being collected (at the age of about 10–11 months). Older, more experienced boars (after six and nine months of service) react more slowly to the phantom, which may be associated with decreasing testosterone concentrations. Some authors reported that higher testosterone is linked to better sperm production and greater libido [4,30].

The increase in the ejaculatory performance of young boars during the first few months of the experimental period when ejaculates were being collected is unsurprising. This response is likely due to the enhanced sexual maturation of the boar during sexual development. As sexual development advances, the ejaculate volume, sperm count in the ejaculate, and number of insemination doses per ejaculate increase, which has also been reported in other studies [31,32,33,34,35].

The ejaculates with the least volume (212.98 mL) were obtained from boars at the start of the experimental period of ejaculate collections. The small ejaculate volume in young boars was also reported in a study by Banaszewska and Kondracki [36], where ejaculate volume of the youngest boars (under 1 year of age), at the start of the interval of ejaculate collections, generally did not exceed 210 mL. The small ejaculate volume in young boars was also reported by Kawęcka et al. [37]. Increasing ejaculation performance with the age of the boar is due to the development of the testicular parenchyma and accessory sex glands [36,38].

According to Savić and Petrović [23], the increase in ejaculate volume with age is due to increased functions of accessory sex glands, particularly the prostate. Other researchers reported that there was no association between ejaculate volume and the age of the boar [39,40]. According to Sharpe et al. [41], the main factor determining sperm production is the number of Sertoli cells. The number of Sertoli cells in the seminiferous tubules of the testes determines the ultimate size and maximum sperm production [42,43].

According to the available literature, the duration of time between the boar’s entrance into the arena and the start of ejaculation may depend on the concentration of sex hormones. Data presented by Louis et al. [44] indicate that the greater the concentration of oestrogens in the blood of the male, the longer the duration is to the beginning of ejaculation. The required duration for ejaculation can be reduced by administering prostaglandins (PGF_2α_) 30 min before collecting the ejaculate [45].

Wysokińska and Kondracki [17] concluded that libido parameters are correlated with ejaculate traits. Results from a study by Xing et al. [46] in White Duroc × Erhualian crossbred boars indicated ejaculate volume was greater in males showing a higher level of sexual activity. This was not confirmed in the present study. There are differing thoughts regarding the existence of an association between values for parameters of sexual activity and the ejaculate traits of boars. Some researchers believe that there is no such association. In a study with White Duroc × Erhualian boars, Ren et al. [47] concluded that values for libido parameters are not correlated with ejaculate traits.

## 5. Conclusions

This study has allowed a number of conclusions to be reached. The sexual experience and libido characteristics of male domestic pigs are not favourable to the expression of sexual behaviours.Young boars in their first three months of use for breeding need less time to begin ejaculation. The ejaculate traits change as the duration of the interval during which semen collection advances. The ejaculates with the most utility were obtained after six months of ejaculate collection; these samples were of the largest volume and contained the most spermatozoa. The time from entering the arena until the start of ejaculation determined at the start of the boar’s use for breeding can be used to predict the extent of libido expression. The boars requiring the longest time to begin ejaculation also had the longest duration of ejaculation and the longest total copulation time, and they mounted the phantom the most times before achieving ejaculation. Finally, there is an association between the extent of a boar’s expression of libido and the ejaculate traits. The boars requiring the longest time to begin ejaculation produced ejaculates with a greater sperm concentration and number.

## Figures and Tables

**Table 1 animals-11-03341-t001:** Time needed to induce copulatory reflexes in boars depending on their copulation experience.

Parameter	Period of Service			
	Start of Service (at the Age of 7–8 Months)	After 3 Months of Service (at the Age of 10–11 Months)	After 6 Months of Service (at the Age of 13–14 Months)	After 9 Months of Service (at the Age of 16–17 Months)
Number of boars	49	49	49	49
Time from entering the arena until mounting the phantom (s)	166.66 ± 161.27	146.55 ± 135.27	154.75 ± 149.76	177.98 ± 163.17
Time from mounting the phantom until achieving erection (s)	43.77 ± 34.97	40.31 ± 25.11	57.47 ± 46.28	61.12 ± 58.60
Time from entering the arena until achieving erection (s)	210.43 ± 166.47	186.86 ± 142.78	212.22 ± 185.54	239.10 ± 182.65
Time from achieving erection until beginning ejaculation (s)	48.04 ± 46.42	58.04 ± 57.46	58.90 ± 46.68	50.06 ± 30.35
Time from entering the arena until beginning ejaculation (s)	258.47 ± 180.85	244.90 ± 171.12	271.12 ± 187.89	289.16 ± 185.99
Duration of ejaculation (s)	397.30 ± 151.72	399.71 ± 199.62	440.24 ± 173.90	430.24 ± 201.00
Total duration of copulation (s)	489.11 ± 194.26	498.06 ± 224.16	556.61 ± 193.17	541.42 ± 205.27
Number of times mounting the phantom	2.20 ± 1.65	1.63 ± 0.92	1.81 ± 1.43	1.55 ± 1.41

**Table 2 animals-11-03341-t002:** Ejaculate traits obtained from boars at the start of insemination service and after 3, 6 and 9 months of semen collection.

Parameter	Period of Service			
	Start of Service (at the Age of 7–8 Months)	After 3 Months of Service (at the Age of 10–11 Months)	After 6 Months of Service (at the Age of 13–14 Months)	After 9 Months of Service (at the Age of 16–17 Months)
Number of boars	49	49	49	49
Ejaculate volume (mL)	212.98 ± 91.49 ^a^	245.92 ± 97.02	255.22 ± 87.98 ^b^	235.88 ± 75.80
Sperm concentration (×10^3^/mm^3^)	399.35 ± 134.35	394.92 ± 166.99	389.10 ± 144.71	384.45 ± 142.95
Percentage of motile sperm (%)	73.77 ± 4.84	73.06 ± 5.08	72.65 ± 4.46	72.45 ± 4.34
Sperm count (×10^9^)	58.18 ± 21.59	62.35 ± 20.55	64.42 ± 19.84	58.43 ± 22.56
Number of insemination doses	19.10 ± 7.05 ^A^	21.92 ± 7.12	23.04 ± 7.21 ^B^	21.73 ± 6.24

Different superscript letters ^a, b^ designate significant differences between values within rows at *p* < 0.05; Different superscript letters ^A, B^ designate significant differences between values within rows at *p* < 0.01.

**Table 3 animals-11-03341-t003:** Results of measurements of sexual activity time required to begin ejaculation at the start of the period of semen collection.

Period of Semen Collection	Group I ≤140 s	Group II 141–290 s	Group III ≥291 s
Number of Boars	16	16	17
Time from entering the arena until beginning ejaculation (s)			
Start of service	96.19 ± 30.13 ^A^	206.25 ± 40.31 ^B^	460.41 ± 152.79 ^C^
After 3 months of service	210.06 ± 199.04	219.62 ± 127.57	301.47 ± 173.78
After 6 months of service	205.56 ± 166.43	300.25 ± 213.38	303.59 ± 177.22
After 9 months of service	278.81 ± 215.16	241.81 ± 105.92	328.70 ± 196.14
Duration of ejaculation (s)			
Start of service	365.94 ± 156.39 ^A^	328.94 ± 131.18 ^A^	491.18 ± 122.39 ^B^
After 3 months of service	420.19 ± 154.79	356.50 ± 125.88	421.12 ± 281.88
After 6 months of service	463.19 ± 164.39	403.44 ± 170.56	443.18 ± 185.10
After 9 months of service	462.87 ± 202.24	416.69 ± 185.93	405.59 ± 222.04
Total duration of copulation (s)			
Start of service	424.31 ± 168.80 ^A^	400.50 ± 126.43 ^A^	633.53 ± 193.15 ^C^
After 3 months of service	497.56 ± 170.03	455.81 ± 127.84	538.29 ± 323.75
After 6 months of service	558.44 ± 183.00	542.37 ± 206.83	553.70 ± 193.51
After 9 months of service	563.87 ± 211.19	530.25 ± 204.08	514.06 ± 206.18
Number of times mounting the phantom			
Start of service	1.37 ± 0.62 ^A^	2.00 ± 0.97 ^a^	3.18 ± 2.29 ^Bb^
After 3 months of service	1.50 ± 0.89	1.62 ± 0.88	1.76 ± 1.03
After 6 months of service	1.50 ± 0.81	2.12 ± 2.10	1.82 ± 0.88
After 9 months of service	1.75 ± 1.23	1.31 ± 0.79	1.47 ± 0.71

Different superscript letters ^a, b^ designate significant differences between values within rows at *p* < 0.05; Different superscript letters ^A, B, C^ designate significant differences between values within rows at *p* < 0.01.

**Table 4 animals-11-03341-t004:** Ejaculate traits depending on the time needed to begin ejaculation.

Period of Semen Collection	Group 1 ≤140 s	Group 2 141–290 s	Group 3 ≥291 s
Number of Boars	16	16	17
Ejaculate volume (mL)			
Start of service	199.25 ± 83.17	199.56 ± 60.08	238.53 ± 119.08
After 3 months of service	248.62 ± 82.42	261.25 ± 102.70	227.76 ± 106.56
After 6 months of service	254.81 ± 76.51	261.37 ± 83.56	249.82 ± 105.45
After 9 months of service	237.06 ± 65.45	241.69 ± 79.54	243.41 ± 106.28
Sperm concentration (×10^3^/mm^3^)			
Start of service	390.93 ± 162.62	406.69 ± 88.74	400.35 ± 147.84
After 3 months of service	397.81 ± 171.87	364.50 ± 135.41	420.82 ± 192.69
After 6 months of service	397.44 ± 146.89	316.56± 110.81 ^A^	449.53± 147.90 ^B^
After 9 months of service	367.69 ± 126.11	366.56 ± 123.99	407.65 ± 173.50
Sperm count (×10^9^)			
Start of service	53.38 ± 24.32	58.24 ± 18.53	62.65 ± 21.88
After 3 months of service	58.21 ± 18.31	64.36 ± 21.19	64.36 ± 22.52
After 6 months of service	60.87± 17.63 ^a^	55.82± 9.16 ^A^	75.86± 24.20 ^Bb^
After 9 months of service	47.92± 11.66 ^A^	55.74 ± 16.08	70.11± 29.78 ^B^

Different superscript letters ^a, b^ designate significant differences between values within rows at *p* < 0.05; Different superscript letters A, B designate significant differences between values within rows at *p* < 0.01.

## Data Availability

No new data were created or analysed in this study. Data sharing is not applicable to this article.

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
