# Peer review of "The Usefulness of Sexual Behaviour Assessment at the Beginning of Service to Predict the Suitability of Boars for Artificial Insemination"

_animals, 2021, doi:10.3390/ani11123341_

Round 1

Reviewer 1 Report

The authors have responded appropriately to the feedback provided by this peer reviewer.

Author Response

Manuscript number: animals-1476501

Answers to the reviewer’s 1 comments

Dear Reviewer,

Thank you very much for reviewing the manuscript entitled „The usefulness of sexual behaviour assessment at the beginning of service to predict the suitability of boars for artificial insemination” (Manuscript ID: animals-1476501) and providing your comments and suggestions. They are reflected in the amended version of the manuscript.

Yours sincerely,

Dr hab. Krzysztof Górski

Reviewer 2 Report

Emendations are appropriate. Now suitable for publication.

Author Response

Manuscript number: animals-1476501

Answers to the reviewer’s 2 comments

Dear Reviewer,

Thank you very much for reviewing the manuscript entitled „The usefulness of sexual behaviour assessment at the beginning of service to predict the suitability of boars for artificial insemination” (Manuscript ID: animals-1476501) and providing your comments and suggestions. They are reflected in the amended version of the manuscript.

Yours sincerely,

Dr hab. Krzysztof Górski

Reviewer 3 Report

The authors extensively revised the paper, which has improved dramatically. They did not, however, answer some of the questions raised and correct the text. In particular,

Line 376 (Table 4 and elsewhere). Physical characteristics. I repeat that sperm concentration, motility and number cannot be included as physical characteristics.

Line 506. I repeat: Are you sure 0.7 mm? I think it is 0.7 µm (700 nm). Visible light waves range between 400 and 700 nm.

It is quite difficult to accurately assess total and progressive sperm motility without diluting the semen.

I did not appreciate the way in which they summarily responded to the criticisms raised by not reporting point-by-point, quoting each criticism and inserting an adequate and comprehensive response.

Line 345. in artificial

Author Response

Manuscript number: animals-1476501

Answers to the reviewer’s 3 comments

Dear Reviewer,

Thank you very much for reviewing the manuscript entitled „The usefulness of sexual behaviour assessment at the beginning of service to predict the suitability of boars for artificial insemination” (Manuscript ID: animals-1476501) and providing your comments and suggestions. They are reflected in the amended version of the manuscript.

Please find attached the revised version of the paper, together with a detailed index of changes made according to the referee’s suggestions.

  • It would have been more effective from a statistical point of view to evaluate correlations between the analyzed parameters or to evaluate the repeatability of this character.

The assessment of the correlation between the analyzed parameters was abandoned due to the high variability of the tested features. Therefore, correlations were considered unreliable.

  • It is unclear how many weekly semen collections the boars were subjected to. Also, all alike, without interruptions?

Now lines: 141-142. During testing of the sexual activity of the boars, ejaculates were collected in the morning twice per week, using the “gloved-hand” technique.

  • Has the semen analyzed to evaluate sperm characteristics been diluted? Which extender was used for this?

The characteristics of the ejaculate were determined on the unsustained semen. During the preparation of insemination doses, the Biosolvens Plus extender was used.

  • Behavioral variations and the quality of the semen associated to the carrying out of the collections have been evaluated; however, considering that the whole experiment took place shortly after the males reached puberty and continued up to the full sexual maturity, biological age would play a decisive role rather than the time of semen collection. In most of the paper, behavioral and semen variations are addressed to the time of semen collection.

Now lines: 121, 122, 123. The biological age of boars at the time of semen collection is given.

  • Have all the semen collections started at the same time or differentiated over time? Were males kept indoors with a controlled light regime or free to go outside? Seasonal variations can significantly affect results.

Now lines: 106, 110. All the boars had the opportunity to use the paddocks. The research began at the same time. The animals were kept in the same conditions.

  • Did the animals belong to a research center or a private company? Are fertility data available?

Now line 104. The study was conducted on 49 boars used at commercial insemination stations.

Sow fertility data has not been checked.

  • Data is missing in the abstract. At least, the main significant differences should be included in it.

Now lines: 41-46 and 48-52. The abstract has been supplemented with the missing data.

  • Line 95. It is not clear: in Poland, there is no need for approval by the Ethics Committee for any type of research conducted on pigs?

Now lines: 101-103. The material and methods section has been supplemented with ethics approval.

  • Throughout the text, the term artificial insemination service is very often used; however, in many cases, it would be more appropriate to replace it with “semen collection”.

Now lines: 45, 121, 122, 123, 126, 127, 145, 205, 212, 213, 220, 223, 228, 232, 233, 236, 241, 243, 248, 261, 264, 265, 267, 269, 283, 284, 298, 304, 330, 389. Term „artificial insemination service” has been replaced with „semen collection”.

  • The authors indicate with "physical characteristics" all seminal and spermatic parameters. This is not appropriate.

Now lines: 147, 154, 227, 231, 276, 279, 283, 397. Term „physical characteristics” has been replaced with „ejaculate traits”.

  • Line 114. delete "by the manual method".

Now line 142. The term "by the manual method" has been removed.

  • Line 134. Replace thousands with 103(as well as in tables and forward).

Now line 149. "thousands" has been replaced with "x 103".

  • Line 137. Replace billions with 109(as well as in tables and forward).

Now lines: 50, 51, 52, 151, 242, 243, 293, 294, 295, 296. „billions” has been replaced with „x 109”.

  • Line 143. SpermaCue photometer (indicate company info).

Now line 157. The sperm concentration in the ejaculate was determined using the photometric method using a SpermaCue photometer (Minitube, GmbH, Tiefenbach, Germany).

  • Line 145. Are the authors sure that the optical path length used was 0.7 mm, or rather nm?

Now line 159. The micro-cuvettes have been developed especially for SpermaCue photometer. The length of the light path is 0.7 mm. (www.minitube.com/userdata/filegallery/original/687_spermnotes-equine_01_2012_en.pdf)

  • Line 166. "... phantom; it was highly variable ..."

Now line 179. Changed as suggested by the reviewer.

  • Line 167. "... differences among groups."

Now line 180. Changed as suggested by the reviewer.

  • Line 169. Replace “To this end”

Now line 181. Replaced as suggested by the reviewer.

  • Lines 169-170. "... according to the sexual activity"

Now line 182. Changed as suggested by the reviewer.

  • Line 171. "... three groups."

Now line 184. Changed as suggested by the reviewer.

  • Line 256. Replace "less pronounced" with "lost".

Now line 275. Terms were changed as suggested by the reviewer.

  • Line 305. "shorter"

Now line 325. Changed as suggested by the reviewer.

  • Line 350. What kind of prostaglandins?

Now line 376. The duration of needed for ejaculation can be reduced by administering prostaglandins (PGF2α) 30 minutes before collecting the ejaculate.

  • Line 368. Replace "link" with "relationship".

Now line 380. Replaced “link” with “association”.

Minor alterations have been implemented taking into account referees’ remarks.

Yours sincerely,

Dr hab. Krzysztof Górski

Round 2

Reviewer 3 Report

The authors have now responded adequately to the criticisms raised and modified the text accordingly. One point remains to be resolved. I have read the brochure of the photometer SDM 1 used for the evaluation of sperm concentration and the text reports as follows, "It measures turbidity with a 546 nm filter".  Then in the description of the characteristics of the cuvettes used it says: "They are precision manufactured. The length of the
light path is 0.7 mm". However, this indication refers only to the cuvettes and not to the wavelength used for the spermatic reading. So, I repeat for the third and last time: the text must be changed.

Author Response

Manuscript number: animals-1476501

Answers to the reviewer’s 3 comments

Dear Reviewer,

Thank you very much for reviewing the manuscript entitled „The usefulness of sexual behaviour assessment at the beginning of service to predict the suitability of boars for artificial insemination” (Manuscript ID: animals-1476501) and providing your comments and suggestions. They are reflected in the amended version of the manuscript.

Please find attached the revised version of the paper, together with a detailed index of changes made according to the referee’s suggestions.

  • Are you sure 0.7 mm? I think it is 0.7 µm (700 nm). Visible light waves range between 400 and 700 nm.

Now line 159: Changed as suggested by the reviewer.

Minor alterations have been implemented taking into account referees’ remarks.

Yours sincerely,

Dr hab. Krzysztof Górski

This manuscript is a resubmission of an earlier submission. The following is a list of the peer review reports and author responses from that submission.

Round 1

Reviewer 1 Report

The research reported in this manuscript was conducted in a manner that the objectives of the study could be effectively addressed. The greatest problem is the written quality of the manuscript is poor. The reviewer has provided extensive editorial feedback on the attached PDF file of this manuscript. If the authors do not effectively address the editorial feedback, this manuscript should not be published.

Author Response

Manuscript number: animals-1405988

Answers to the reviewer’s 1 comments

Dear Reviewer,

Thank you very much for reviewing the manuscript entitled „The usefulness of assessment of sexual behaviour at the start of service to predict the suitability of boars for artificial insemination” (Manuscript ID: animals-1405988) and providing your comments and suggestions. They are reflected in the amended version of the manuscript.

Manusctipt is after English Editing Service from MDPI. Please find attached the revised version of the paper, together with a detailed index of changes made according to the referee’s suggestions.

The manuscript has been redrafted using the comments of the reviewer 1.

Minor alterations have been implemented taking into account referees’ remarks.

Sincerely,

Krzysztof Górski

Reviewer 2 Report

This article describes investigations into the relationship between sexual behaviour and potential contribution to an AI program of boars. While the topic is of interest and the article well written, there are several serious flaws that lead me to recommend major revision. The attached pdf with inserted comments gives more detail, but the major concerns were:

1) inadequate ethics approval for the study. However I understand the different standards are acceptable in different countries and so I look to the editor to decide on this matter.

2) inadequate description of methods - many factors that may influence boar sexual behaviour were not outlined (examples of these are given in the attachment). Even factors such as season mentioned by authors in the discussion were not mentioned in the methods.

3) discussion was missing important areas. as outlined in the attached document but also around the fact that the data in Table 4 showed the boars with an intermediate time to ejaculation were on occasion lower than both the shorter and longer time to ejaculation groups. Additionally, I think that conclusions were not adequately justified.

Author Response

Manuscript number: animals-1405988

Answers to the reviewer’s 2 comments

Dear Reviewer,

Thank you very much for reviewing the manuscript entitled „The usefulness of assessment of sexual behaviour at the start of service to predict the suitability of boars for artificial insemination” (Manuscript ID: animals-1405988) and providing your comments and suggestions. They are reflected in the amended version of the manuscript.

Manusctipt is after English Editing Service from MDPI. Please find attached the revised version of the paper, together with a detailed index of changes made according to the referee’s suggestions.

  • The „Material and methods” section has been supplemented with ethics approval.
  • The „Material and methods” section has been supplemented as suggested by the reviewer with the missing description of methods.
  • The „Discussion” section has been supplemented as suggested by the reviewer with the views of other authors regarding testosterone concentration.

Minor alterations have been implemented taking into account referees’ remarks.

Sincerely,

Krzysztof Górski

Reviewer 3 Report

The paper is clearly written, the experimental design is well constructed even if it could have been thorough, the results are clear as well as the conclusions. There are, however, several parts that need to be clarified and modified to improve the quality of the paper and allow it to be published.

Particularly:

The authors, after a preliminary examination characterized by high variability of the results, decided to divide the boars, among the various parameters analyzed, in relation to the time taken between entering the arena and the onset of ejaculation. It would have been more effective from a statistical point of view to evaluate correlations between the analyzed parameters or to evaluate the repeatability of this character.

It is unclear how many weekly semen collections the boars were subjected to. Also, all alike, without interruptions?

Has the semen analyzed to evaluate sperm characteristics been diluted? Which extender was used for this?

Behavioral variations and the quality of the semen associated to the carrying out of the collections have been evaluated; however, considering that the whole experiment took place shortly after the males reached puberty and continued up to the full sexual maturity, biological age would play a decisive role rather than the time of semen collection. In most of the paper, behavioral and semen variations are addressed to the time of semen collection.

Have all the semen collections started at the same time or differentiated over time? Were males kept indoors with a controlled light regime or free to go outside? Seasonal variations can significantly affect results.

Did the animals belong to a research center or a private company? Are fertility data available?

Minor criticisms

Data is missing in the abstract. At least, the main significant differences should be included in it.

Line 95. It is not clear: in Poland, there is no need for approval by the Ethics Committee for any type of research conducted on pigs?

Throughout the text, the term artificial insemination service is very often used; however, in many cases, it would be more appropriate to replace it with “semen collection”.

The authors indicate with "physical characteristics" all seminal and spermatic parameters. This is not appropriate.

Line 114 delete "by the manual method"

Line 134. Replace thousands with 103 (as well as in tables and forward)

Line 137. Replace billions with 109 (as well as in tables and forward).

Line 143. Sperma Cue photometer (indicate company info)

Line 145. Are the authors sure that the optical path length used was 0.7 mm, or rather nm?

Line 166; "... phantom; it was highly variable ..."

Line 167. "... differences among groups."

Line 169. Replace “To this end”

Lines 169-170. "... according to the sexual activity"

Line 171. "... three groups."

Line 256. Replace "less pronounced" with "lost".

Line 305. "shorter"

Line 350. What kind of prostaglandins?

Line 355. "... is, however, controversial; in fact, other studies (Ren et al., 2009) did not find any relationship between these parameters.

Line 368. Replace "link" with "relationship".

Author Response

Manuscript number: animals-1405988

Answers to the reviewer’s 3 comments

Dear Reviewer,

Thank you very much for reviewing the manuscript entitled „The usefulness of assessment of sexual behaviour at the start of service to predict the suitability of boars for artificial insemination” (Manuscript ID: animals-1405988) and providing your comments and suggestions. They are reflected in the amended version of the manuscript.

Manusctipt is after English Editing Service from MDPI. Please find attached the revised version of the paper, together with a detailed index of changes made according to the referee’s suggestions.

  • The assessment of the correlation between the analyzed parameters was abandonem due to the high variability of the tested features. Therefore, correlations were considered unreliable.
  • Ejaculates were collected twice per week.
  • The ejaculate traits were determined on undiluted, fresh semen. During the preparation of insemination doses, the Biosolwens Plus extender was used.
  • All the boars had the opportunity to use the paddocks. The research began at the same time. The animals were kept in the same conditions.
  • The „Material and methods” section and Tables have been supplemented as suggested by the reviewer with the missing description of boars age.
  • The abstract has been supplemented with the missing data.

Minor alterations have been implemented taking into account referees’ remarks.

Sincerely,

Krzysztof Górski